# Managing Practical Resistance of Lepidopteran Pests to Bt Cotton in China

**DOI:** 10.3390/insects14020179

**Published:** 2023-02-10

**Authors:** Yudong Quan, Kongming Wu

**Affiliations:** 1State Key Laboratory for Biology of Plant Diseases and Insect Pests, Institute of Plant Protection, Chinese Academy of Agricultural Sciences, Beijing 100193, China; 2Guangdong Laboratory for Lingnan Modern Agriculture, Guangzhou 510641, China

**Keywords:** Bt cotton, *Helicoverpa armigera*, *Pectinophora gossypiella*, refuge, hybrid F2 seed, IRM, pyramided strategy, resistance monitoring, pest management

## Abstract

**Simple Summary:**

Bt (Cry1Ac) cotton has been commercialized in China since 1997. Two tactics have mainly been used in targeted pest resistance management for small farmers. The first is the use of natural refuges composed of corn, soybeans, vegetables, peanuts, and other crops to delay the development of the resistance of multi-host crop pests such as the cotton bollworm (*Helicoverpa armigera*) to Bt cotton. The second is to use the offspring of F2 generation hybrids (Bt and non-Bt cotton hybrids) to produce a 25% seed mix refuge formed by non-Bt cotton, further delaying the evolution of resistance of single cotton-eating target pests such as the pink bollworm (*Pectinophora gossypiella*). The 25-year production practice has demonstrated that this dual strategy of Bt cotton resistance management in China has been very successful to date, with no failures observed in lepidopteran pest control in the production process.

**Abstract:**

China is one of the major cotton producers globally with small farmers. Lepidopteran pests have always been the main factor affecting cotton production. To reduce the occurrence of and damage caused by lepidopteran pests, China has employed a pest control method focused on planting Bt (Cry1Ac) cotton since 1997. Chinese resistance management tactics for the main target pests, the cotton bollworm and pink bollworm, were also implemented. For polyphagous (multiple hosts) and migratory pests such as the cotton bollworm (*Helicoverpa armigera*), the “natural refuge” strategy, consisting of non-Bt crops such as corn, soybean, vegetables, peanuts, and other host crops, was adopted in the Yellow River Region (YRR) and Northwest Region (NR). For a single host and weak migration ability pest, such as the pink bollworm (*Pectinophora gossypiella*), the seed mix refuge strategy yields a random mixture within fields of 25% non-Bt cotton by sowing second-generation (F2) seeds. According to field monitoring results for more than 20 years in China, practical resistance (Bt cotton failure) of target pests was avoided, and there were no cases of Bt (Cry1Ac) failure of pest control in cotton production. This indicated that this Chinese resistance management strategy was very successful. The Chinese government has decided to commercialize Bt corn, which will inevitably reduce the role of natural refuges; therefore, this paper also discusses adjustments and future directions of cotton pest resistance management strategies.

## 1. Introduction

Cotton is an important source of fiber for the textile industry and biofuels (seed oils) and is a major annual cash crop. Globally, about 150 countries are involved in the cotton industry, with an annual output value of more than USD 60 billion [1,2]. Outbreaks of cotton pests have directly or indirectly (transmission of plant diseases) caused 15–30% economic losses during production [3,4,5]. An attempt to reduce the damage has led to the overuse and misuse of chemical insecticides, leading to serious health and ecological problems, including harm to human and animal health, insecticidal resistance, pest resurgence, and environmental pollution [6,7]. Therefore, green and efficient control of cotton pests is of great significance to maintaining the healthy development of the industry.

Lepidoptera includes the most important taxa of cotton pests globally [8]. Since the 1980s, *Bacillus thuringiensis* (Bt) (or secreted protein) has been widely used as a microbial insecticide to control cotton pests such as the cotton bollworm (*Helicoverpa armigera*), pink bollworm (*Pectinophora gossypiella*), and beet armyworm (*Spodoptera exigua*). With the development of biotechnology, genetically engineered crops that produce insecticidal Bt proteins in plant tissue were created that can kill some voracious insect pests but are not toxic to most non-target organisms, including natural enemies. Bt cotton was commercialized in the United States and Australia in 1996 [9,10]. Since then, China (1997), Argentina (1998), South Africa (1998), Colombia (2002), India (2002), and Brazil (2005) have successively approved the production and application of Bt cotton [11,12]. At present, the insecticidal protein species expressed by commercially cultivated Bt cotton around the world include Cry1Ac, Cry2Ab, and Vip3Aa, with an annual planting area of more than 33 million hectares [2,5,13]. This plays an important role in controlling the occurrence of lepidopteran pests (see Tabashnik and Carrière (2019) [14], Li et al., (2019) [15], Knight et al., (2021) [16] review).

Despite the widespread application of Bt cotton suppressing targeted pests, the primary threat to its long-term efficacy is evolution of resistance by pests, which entails a genetically based decrease in their susceptibility that would diminish the benefits of Bt cotton. The preventive management of resistance is an important task for commercial cultivation in various countries. In view of the type of Bt cotton (differently expressed Bt proteins), cultivation, target pests, and other factors, various countries have formulated their own resistance management strategies based on local situations [17]. In China, cotton cultivation and production are mainly performed by millions of smallholder farmers. The occurrence of pests in large-scale planting regions is very different due to the climate [8,18]. Bt cotton producing Cry1Ac was commercialized about 25 years ago in China and has been effective against the cotton bollworm (*H. armigera)*, a serious pest of many crops that is unique to cotton-growing countries. In this paper, we review the commercialization process, resistance management strategies, and effectiveness of Bt-Cry1Ac cotton in China in order to enable scientists from all over the world to better understand China’s experience, enrich the understanding for resistance management strategies of smallholder farmers in developing countries, and further promote the sustainable use of Bt crops.

## 2. Bt Cotton Planting and the Target Pest Resistance Management Strategy in China

At present, China’s cotton growing area can be grouped into three major regions: the Yellow River Region (YRR) (33′ N and 41′ N, with an average annual rainfall of 500 to 700 mm), the Yangtze River Region (CRR) (25′ N and 33′ N, with an annual rainfall of 800 to 1500 mm), and the Northwest Region (NR) (Gansu Province, Xinjiang, with an annual rainfall of less than 200 mm) [8]. Before 2000, cotton in China was mainly planted in the YRR and CRR. Since the 21st century, with the adjustment of China’s planting structure, the cotton planting area has gradually shifted to the northwestern inland cotton area. In 2021, the cotton planting area in Xinjiang was 2.5 million hectares, accounting for 82.76% of the total area of the country (https://www.chyxx.com/industry/202112/990259.html, accessed on 29 December 2022). The important Lepidopteran pests in cotton fields in China include the cotton bollworm (*H. armigera),* pink bollworm (*P. gossypiella*)*,* Asian corn borer (*Ostrinia furnacalis*)*,* beet armyworm (*Spodoptera exigua*)*,* tobacco cutworm (*Spodoptera litura*)*,* diamond bollworm (*Earias cubeoviridis*)*,* and cotton looper (*Anomis flava*). Traditionally, the cotton bollworm is the most important pest of cotton in the YRR and NR, while the pink bollworm is the most important pest of cotton production in the CRR [8].

In response to the serious threat to cotton production from lepidopteran pests such as the cotton bollworm and pink bollworm, China approved the commercial cultivation of cotton varieties expressing Cry1Ac in YRR in 1997, which was expanded to CRR areas after 2000 and was also widely cultivated in Xinjiang and Gansu after 2009 [18,19,20]. In the early stage, Bt cotton varieties grown in China were mainly from Monsanto and were gradually replaced by domestic cotton varieties (GK series) [21,22]. Bt (CryAc) cotton has a control effect on a variety of lepidopteran pests. Its control effect on pests such as the pink bollworm and Asian corn borer is greater than 90%, and its control effect on pests such as the cotton bollworm is higher than 80%. Its control effect on the noctuid twill moth and beet armyworm moth, however, is relatively low [23,24,25,26,27,28]. After the large-scale planting of Bt cotton, the incidence of lepidopteran pests in various cotton areas was significantly reduced, meaning the use of chemical pesticides was basically unnecessary. This played a critical role in the development of cotton production and was generally welcomed by cotton growers [13,18,29,30].

The widespread and long-term use of Bt cotton has greatly increased the risk of generating resistance to the Cry1A toxin by target insects in the field [8]. It is agreed that protoxin (the full-length Cry1Ac protein) is digested by midgut proteases (of target insects) into activated toxins, then the activation would bind to insect midgut receptors and induce the formation of pores in the membrane thus leading to cell death [31]. Since the highest changes in activity have been detected through alterations in binding to midgut receptors, this step is considered critical [31]. Diverse proteins and glycoconjugates contain binding sites of *H. armigera* and *P. gossypiella* in the field or laboratory population in China, such as cadherin, teraspanin, aminopeptidase-N (APN), alkaline phosphatase (ALP), and ATP-binding cassette (ABC) transporter family proteins (Table 1). Such proteins have been proposed to explain resistance to Cry toxins. In addition, given that there is no shared binding site (receptor) between Vip3 and Cry proteins, their use in combination favors the toxicity of Bt crops and also delays insect resistance [17].

The resistance of target insects can generally be classified into three previously described categories: practical resistance, early warning of resistance (the resistance has significantly increased, but the Bt crops can still control or kill most populations), and no decrease in susceptibility (no significant increase compared with the baseline) [31]. To delay the resistance in the field, several resistance management strategies have been developed, including “high-dose refuge”, “multiple genes” (such as *Cry* and *Vip3*) (pyramid strategy), and rotation of Bt crops [17]. Given the large holdings and relatively homogenous cropping patterns in developed countries such as the USA and Australia, the use of mandated, structural refuges is feasible. However, due to small land holdings and heterogeneous cropping patterns in developing countries such as India and China, the use of structural refuges is difficult. Although the Chinese government has put forward requirements, implementing and supervising compatible refuges remains challenging [32,33,34]. According to China’s conditions, two special resistance management strategies can be used. One is called “natural refuges” and is composed of other non-Bt crops (hosts) such as corn, soybean, peanuts, etc., planted around Bt cotton, to delay the resistance of the polyphagous cotton bollworm. The other is for the pink bollworm and is called “seed mix refuges”. It is established through planting F2 hybrid cotton seeds generated by the hybridization of Bt cotton and ordinary cotton [23,35]. After about 20 years, different effects of these resistance management strategies can be observed. The resistance management measures in developed countries such as the United States have effectively controlled the development of cotton pest resistance, while the resistance management measures in China have also been successful; however, in countries such as India, there have been failures in Bt cotton pest control.

**Table 1 insects-14-00179-t001:** The reported resistance mechanism of *H. armigera* and *P. gossypiella* in previous publications in China.

Insects	Resistance Mechanism			Inheritance	Reference
	Mutations	Genes	Frequency (%)		
*H. armigera*	Protease	*Haserpin-e* (up)	/ ^2^	/	[36]
		*HaTryR* (down)	/	/	[37]
	Cadherin ^1^	*HaCad* (down)	8.7%	Recessive	[24,31]
	Tetraspanin ^1^	*HaTSPAN1* (down)	10%	Dominant	[38]
	ATP-binding cassette transporters (ABCC2/3) ^1^	*HaABCC2/3* (down)	/	Recessive	[39]
	Aminopeptidase N (HaAPN1-96S)	*apn1*	/	/	[40]
	Alkaline phosphatase (HaALP1f)	*HaALP1/2*	/	Matrilineal	[41]
*P. gossypiella*	Cadherin ^1^	*r1/2PgCad1*	*/*	Recessive	[42]
		*r13PgCad1*	/	Recessive	[43]
		*r14PgCad1*	/	Recessive	[44]
		*r15PgCad1*	*/*	Recessive	[45]
		*r16PgCad1*	*/*	Recessive	[46]
		*r17/18PgCad1*	/	Recessive	[42]
		*r19/20PgCad1*	/	Recessive	[47]
		*PgCad189/88*	0.02/3%	Recessive	[48]
		*PgCad47*	0.03%	Recessive	[48]
		*PgCad65*	/	Recessive	[48]

“^1^” Found in a field population; the others were detected in the selected Cry1Ac resistance strain in the laboratory. “^2^” Lack of data.

## 3. Resistance Management Tactic Type I: Natural Refuge for Polyphagous Cotton Bollworm

The outbreak of the cotton bollworm in China in the 1990s was the greatest factor restricting cotton production in China [49,50]. Cotton bollworms in China include four geographical types (tropical, subtropical, temperate, and Xinjiang), which are distributed in the tropical region, CRR, YRR, and the Xinjiang Uyghur Autonomous Region of China. In the tropical region and CRR, due to heavy rainfall in the summer, cotton bollworms find it difficult to pupate in the soil, meaning they cause greater harm to cotton production only in dry years. YRR and Xinjiang are major occurrence areas of cotton bollworm in China. The cotton bollworm has a strong migration habit, and each geographical type migrates between host crops in its distribution area [8]. In the YRR and Xinjiang region, the first generation of cotton bollworm larvae feed mainly on wheat, whereas the subsequent generations feed on a variety of crops such as cotton, corn, vegetables, peanuts, and soybeans [8,18]. Due to the small planting plots and the large number of smallholder farms, supervision is difficult, so it is extremely challenging to set up large-scale artificial structured refuges in China. Two methods are highlighted in the management of cotton bollworm resistance to Bt cotton. The first is to plant Bt cotton varieties expressing high concentrations of insecticidal toxin, and the second is to utilize the natural ecological role of small-scale farmers in multi-crop cultivation [18,35]. Generally, cotton is planted together with corn, peanuts, soybeans, and other crops in China. The overlap of the silk/kernel stage of corn and the migration of adult cotton bollworms increases the probability of moth movement and oviposition in corn fields [8]. The first-generation larvae primarily feed on wheat hosts and subsequently migrate to Bt cotton, corn, peanuts, and soybeans (Figure 1) [8,35]. Since crops such as wheat, corn, peanuts, soybeans, and vegetables in the field do not express the Bt protein, they can provide a natural refuge for cotton bollworms that transfer to these hosts. Such crops can produce a large quantity of cotton-bollworm-sensitive individuals (SS), and resistant individuals (RR) living in Bt cotton produce heterozygotes (RS) after mating with sensitive individuals (SS). Given that the inheritance of cotton bollworm resistance to Cry1Ac (expressed in Bt cotton) is autosomal recessive, the heterozygous (RS) resistance allele is not expressed or shows little expression (Table 1). The heterozygous (RS) individuals cannot survive on Bt cotton, thereby reducing the allele frequency (Figure 1). This mixed cropping system, consisting of wheat, corn, soybeans, etc., can provide a refuge for cotton bollworms throughout the season and has been used for the resistance management of pests such as the cotton bollworm in YRR and Xinjiang cotton-growing areas in China [8,35]. At the same time, the long-distance migration of adult cotton bollworms in cotton-growing areas and non-cotton-growing areas in China has also intensified gene exchange between different populations, further diluting the resistance allele frequency [51,52,53,54].

Resistance monitoring is essential for the early detection of Bt resistance, and it is necessary to diagnose the evolutionary status of resistance to the Bt toxin in target insect populations [8,55]. The Chinese government established a national cotton pest prediction and monitoring system in the 1980s and included target pest resistance monitoring after Bt cotton was grown commercially [8,56,57]. Zhang et al. monitored cotton bollworm resistance in China from 1997 to 2017, and the results showed that the IC50s values were less than 0.05 ug/mL before 2014, increased in the subsequent time, and reached 0.092 ug/mL in 2017 (Figure 1B) [24]. F1 generation resistance surveillance methods showed that the frequency of resistant Cry1Ac alleles in field cotton bollworm populations increased from 0.58% to 7.5% (1999–2007), and the frequency of resistance genes fluctuated until 2015 but remained below about 10% without a significant increase (Figure 1C) [58,59,60]. However, in the Xinjiang region (NR), alleles have remained consistently low (under 0.4%) [61]. In addition, monitoring results (2002–2008) from Xiajin County (an intensive Bt cotton-planting area) in the Shandong Province and Anci County in the Hebei Province (multiple crop systems including corn, soybean, peanuts, and Bt cotton) showed that the resistance evolution rate of the cotton bollworm with natural refuges (Anci County) was significantly lower than that in dense cultivation areas (Xiajin), which verified the effectiveness of resistance management strategies in natural refuges [58]. Extensive monitoring of Bt cotton (YRR) in six northern provinces in China from 2010 to 2013 by Jin et al. also indicated that natural refuges could delay the development of cotton bollworm resistance but were less effective than non-Bt cotton-structured refuges in the same area [62]. Li et al. summarized three elements of the efficient role of natural refuges. One is that there are enough host plants in space and time around non-Bt crops to provide living conditions for target insects, another is that these non-Bt crops can produce sufficient freely moving sensitive insects, and the third is that the viable population of Bt crops can freely mate with the natural refuge populations [63].

**Figure 1 insects-14-00179-f001:**
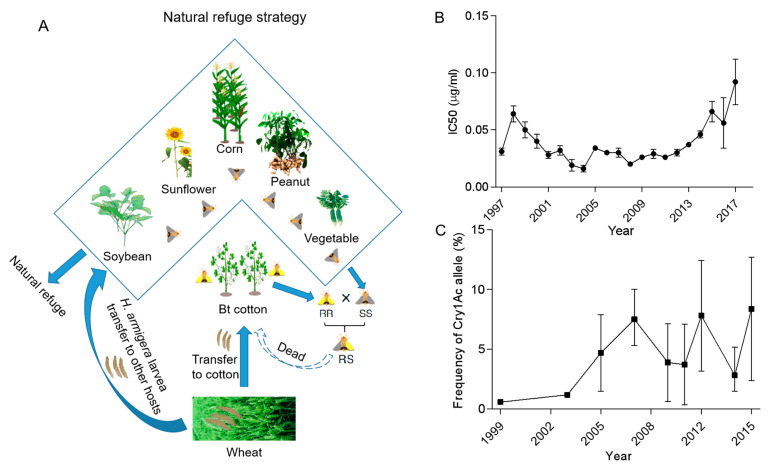
Natural refuge strategy (**A**) and field monitoring of *Helicoverpa armigera* regarding Cry1Ac resistance IC50s and alleles (**B**,**C**) in China. The IC50 is the concentration of the Cry1Ac protein that inhibits larval development by 50%, and the error bars stand for the 95% fiducial limits. The frequency of alleles in field *Helicoverpa armigera* was obtained through an F1 screen from the previous work. Bioassay data of IC50 and frequency of alleles in the figures are from Zhang et al. [24,56,57].

The resistance management of cotton bollworm pests in developed countries such as the United States and Australia includes three measures. One is to plant 5–20% non-Bt cotton around Bt cotton-planting areas and to build susceptible cotton bollworm refuges. It is forbidden to spray Bt pesticides in refuges, and the aim is to kill overwintering diapause pupae as much as possible in Bt cotton fields [32,33]. The second is to plant Bt cotton varieties expressing high concentrations of insecticidal toxins. The third is to plant Bt cotton varieties (such as Cry1Ac + Cry2Ab + Vip3Aa) that express multiple insecticidal genes [64,65]. The implementation of these measures has played a role in adversarial governance, and so far, *H. punctigera* and *H. armigera* in Australia, in addition to *H. virescens* in the United States, remain sensitive to Bt cotton [14]. In contrast, cotton bollworm resistance management measures in China are based on natural refuge production models, which are simpler to operate and more practical to use and are therefore easier for smallholder farmers in developing countries to implement.

## 4. Resistance Management Tactic Type II: Seed Mix Refuge for Oligophagic Pink Bollworm

The pink bollworm (*P. gossypiella*) has been found in various cotton regions in China, spreading to the Yangtze River Basin Cotton Region (CRR) [8,66]. Unlike the cotton bollworm, which has a variety of host crops other than cotton, the pink bollworm mainly feeds on cotton. The first generation of larvae feeds on cotton buds and flowers, causing shedding, and the subsequent generations of larvae drill into the interior of the cotton bolls to feed and damage them, resulting in rotten bolls and stiff petals. Although pink bollworm larvae can feed on okra (*Abelmoschus esculentus*) and tomato (*Lycopersicon esculentum*), field trial investigations have shown that okra or tomato are non-preferred hosts. The pink bollworm survives the winter as larvae in seed cotton, cottonseed, and dead bolls and has 3–4 generations per year in the CRR cotton area [23,67,68].

Regarding the resistance management strategy for the pink bollworm, the United States adopted a structured refuge/high-does approach from 1996 to 2005 and planted 25% non-Bt cotton as a structured refuge every year [69]. Beginning in 2006, cotton growers collaborated with scientists from the U.S. Department of Agriculture, the University of Arizona, and other institutions and released sterile (irradiated) pink bollworm moths in large numbers in the southwestern United States in an effort to eradicate the pink bollworm completely. Sterile moths mated with potentially resistant pink bollworm moths (on Bt cotton), producing relatively few heterozygous larvae (pink bollworms are incompletely dominant for Cry1Ac/Cry2Ab resistance or recessive) which could not survive on Bt cotton. In addition, sterile moths also interacted with wild-type moths in the field (field sensitive or resistant), and since one of the gametes was sterile, these heterozygous larvae were also less likely to become fertile adults. During this period, growers were allowed to grow up to 100% Bt cotton under a special EPA exemption, and the percentage of cotton grown in Arizona that consisted of non-Bt cotton refuges dropped from >25% to less than 5%. Subsequently, annual monitoring in Arizona showed that no pink bollworm larvae were found on cotton from 2010 to 2018, and no wild pink bollworm moths were found in fields from 2013 to 2018. Based on similar conditions in other states, the US Secretary of Agriculture announced in October 2018 that pink bollworm had been eradicated in the continental commercial cotton producing areas in the United States [16,70,71,72,73,74,75]. In India, the cotton seeds provided by cotton seed companies not only include 450 g of Bt cotton seeds per bag but also contain 120 g of non-Bt cotton seeds in order to set up structural refuges. However, farmers have found that non-Bt cotton is harmful to pests, and they are unwilling to plant it, resulting in very few non-Bt cotton refuges in actual production [76]. In 2008, Gujarat confirmed for the first time that pink bollworms had generated resistance to Cry1Ac cotton; thus, Cry1Ac + Cry2Ab cotton had to be planted [77,78]. By 2015, pink bollworm resistance to Cry1Ac + Cry2Ab cotton had become ubiquitous [77,78], indicating that it is difficult for smallholders in developing countries to manage resistance with structural refuge strategies under their production models.

Different strategies have been adopted for the management of resistance of the pink bollworm to Bt cotton in China. In the early stage (2004–2009), smallholders were unwilling to set up structural refuges, leading to the proportion of non-Bt cotton planted being as low as 6% [23]. The concept of natural refuges does not apply to the pink bollworm, which feeds almost exclusively on cotton in the CRR (there are no suitable non-Bt crop refuges). Bt (CryAc) cotton has high control efficiency against the pink bollworm, and the habit of the pink bollworm rarely transplants damage, which provides feasibility for the method of mixed planting with non-Bt cotton and Bt cotton seeds [23,79]. Compared with setting up structural refuges, this hybrid strategy is simpler and easier to carry out. In China, farmers prefer to plant hybrid cotton that combines Bt and non-Bt cotton because it gives a higher yield. Most farmers choose to grow F2 seeds due to the high price of F1 seeds. However, F2 seeds can produce 25% refugia of non-Bt plants (randomly scattered in Bt cotton fields) (Figure 2) [23,80,81,82]. The production process of F2 hybrid cotton seeds involves F1 hybrids (which can produce Bt toxin), crossing Bt cotton with non-Bt cotton (which is more productive than Bt cotton), and F1 generation self-crossing (self-pollination) to produce F2 seeds. Since the F1 generation has one copy of the Bt transgene without corresponding alleles, F1 can still express the Bt toxin. Among F2 seeds produced by self-pollination, 25% Bt homozygotes and 50% heterozygotes are expected to produce Bt toxin, and 25% of non-Bt homozygotes do not develop Bt toxicity [83,84]. The 25% non-Bt cotton provides a large enough sensitive population (SS) for hybridization (RS) with the resistant population (RR) surviving on Bt cotton. Given that the inheritance of Cry1Ac-resistance in the pink bollworm is recessive, heterozygotes (RS) cannot survive on Bt cotton planted in large quantities in the field, reducing the frequency of resistant alleles (Table 1). In terms of seed production, the production of F1 hybrid seeds requires expensive artificial pollination, while the self-pollination of F1 hybrids to produce F2 hybrid seeds can significantly reduce the cost of seed production (the price of F1 seed is about 35% higher than that of F2), and the yield of hybrid cotton (F1/F2) is usually higher than that of its parental varieties (such as parental Bt cotton), further increasing production efficiency [23,81,82]. In 2017, Wan and other scientists tested about 14,000 seeds among 84 of the most prevalent species in the CRR in the Chinese market using an immunoassay to estimate the percentage of seeds containing Cry1Ac in cotton. After analyzing the results, together with planting data in the region, they found that the cotton species in the market from 2004 to 2009 were mainly F1 hybrid cotton (>60%) with only a small amount of F2 hybrid cotton (about 16%), and the subsequent (2010–present) F2 hybrid cotton planting increased to 59% and remained stable [23]. About 15% (59% * 25% ≈ 15%) of the cotton in the 2010 F2 blend field was non-Bt cotton (refuge). Combining non-Bt cotton varieties (small numbers of non-Bt cotton planted by some farmers) and non-Bt cotton plants of F2 hybrids, the overall proportion of refuge (the proportion of non-Bt cotton planted) almost doubled from 12% in 2009 to 23% in 2010 and then increased to 25–27% in 2011–2015 [23].

The monitoring data of pink bollworm resistance in the CRR cotton area showed that when F2 cotton seeds were not planted on a large scale (mainly F1 cotton seeds were planted) from 2004 to 2009, under the diagnostic concentration of 9 μg/mL Cry1Ac, the survival rate of pink bollworm larvae (with newly hatched larvae as the experimental object) increased from 0% in 2005–2007 to 56% in 2008–2010. The median survival rate grew from 0% in 2005–2007 to 1.6% in 2008–2010. The average LC50 (median lethal concentration) (to Cry1Ac) value of pink bollworms collected and monitored in the field increased by about one fold in 2008–2010 compared with that in 2005–2007 (Figure 2A,B) [23,85]. However, after the implementation of seed mix refuges, the percentage of the population surviving at the diagnostic concentration dropped from 56% in 2008–2010 to 0% in 2011–2015, and the median percentage surviving at the diagnostic concentration decreased from 1.6% in 2008–2010 to 0% in 2011–2015. The average LC50 value was 0.26μg/mL in 2012–2015, which decreased to the same as that in 2005–2008 (0.22 μg/mL) (Figure 2D) [23]. In addition, the frequency of resistance alleles also decreased significantly during 2012–2015 [23], which demonstrated that China achieved success in the resistance management of F2 hybrid-based seed mix refuges to the pink bollworm [23]. Hybrid refuges represent an alternative strategy to lower the risk of smallholders not planting structural refuges that meet the requirements. China’s F2 generation hybrid seed strategy has increased farmers’ enthusiasm for implementation by reducing production costs.

## 5. Challenges and Directions of Cotton Pest Resistance Management Strategies in China

The planting of Bt cotton in the past two decades has brought tremendous economic and ecological benefits to Chinese farmers. The application of natural refuges and F2 hybrid seed strategies has played an important role in the long-term use of Bt cotton. However, the deeper commercialization of Bt corn and Bt soybean in China will pose a challenge to the natural refuge strategy of the cotton bollworm [86]. Small-scale experiments have shown that the control effect of transgenic insect-resistant corn on the fall armyworm (*Spodoptera frugiperda)* could reach 95%, and the yield could increase by 6.7–10.7%, thus greatly reducing the cost of pest control [87]. The increase in yield and reduction in control costs may attract a large number of Chinese farmers to widely plant related varieties. Insect-resistant corn varieties that have been issued safety certificates mainly express proteins such as Cry1Ab, Cry1Ab/Vip3Aa, Cry1Ab/Cry2Aj, and Cry1Ab/Cry2Ab, etc., and they efficiently target pests such as *H. armigera*, *O. furnacalis*, *S. frugiperda,* and *Mythimna separata* (lepidopteran pests) [15,88]. Considering that corn and soybean are important natural refuges (non-Bt crops) for resistance management of the cotton bollworm and that >55% of the farmland planting area (in the YRR/CRR area) has corn as a major crop, it is expected that a large amount of Bt corn may replace the original non-transgenic corn hectares. Thus, resistance of the cotton bollworm and other pests of cotton and corn could increase quickly, which will require the rapid development of new resistance management strategies given the complex interactions of multiple insect-resistance genes, regional planting refuge crops, and pest ecology. One caveat to the concerns for resistance is that the addition of Bt corn to production systems in the U.S. has led to significant levels of pest suppression in both non-Bt corn and non-Bt vegetable crops, which has reduced crop losses in the refuge crops while still providing non-Bt-exposed susceptible moths for resistance management [89,90]. In addition, Bt corn planting will also affect the number of natural enemies and reservoirs of cotton pests, which are related to pest population evolution and resistance management in Bt cotton fields. This needs to be further studied.

The strategy of stacking multiple insect-resistant genes (multi-Bt gene cotton), called second-/third-generation cotton (such as Cry1Ac + Cry2Ab + Vip3Aa), has the advantages of a more robust, long-term efficacy and resistance management impact as well as a requirement for fewer refuges compared to monogenic cotton [91]. After 2003, the United States and Australia almost completely replaced single-gene cotton (Cry1Ac, INGARD^®^) with double-gene cotton (Cry1Ac + Cry2Ab, Bollgard II^®^). Subsequently, cotton expressing three genes (Cry1Ac + Cry2Ab + Vip3Aa) was used to replace the double-gene cotton. According to statistics, third-generation cotton expressing Cry and Vip3Aa planted in the United States in 2019 accounted for about 27% of the total planting area, and more than 90% of the planting area in Australia was third-generation cotton during 2016–2017 [5,13,92]. China’s cotton area is still dominated by single-gene Bt cotton at present, and progress of the application of new-generation cotton is far behind that of other countries. Since the F2 generation hybrid seed strategy can provide 25% non-Bt cotton refuges, the expansion of planting non-Bt cotton refuges is not only of great significance for the control of the pink bollworm but can also effectively alleviate the reduction of natural refuges for cotton bollworms affected by the commercial planting of Bt corn and soybeans. Therefore, the research and development of multiple Bt gene cotton, combined with an F2 generation hybrid seed strategy, may be one of the key development directions for small farmers of China in the near future.

## Figures and Tables

**Figure 2 insects-14-00179-f002:**
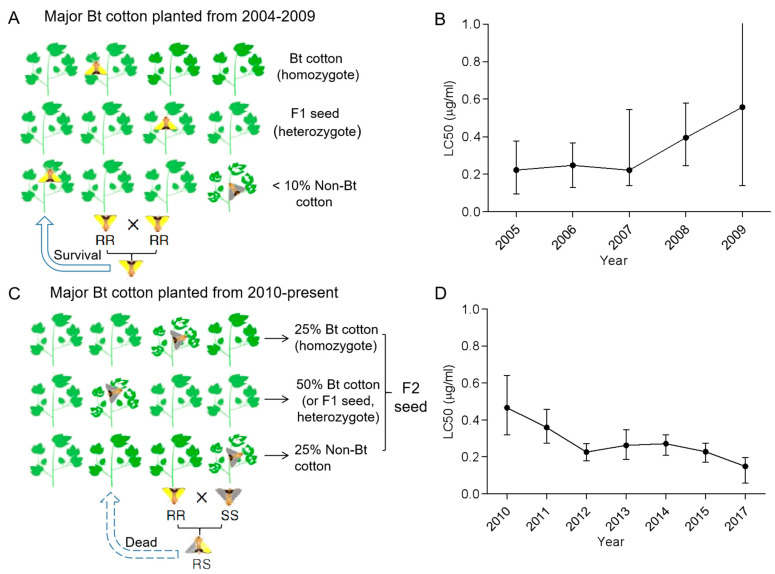
Major Bt cotton planted (**A**,**B**) and field monitoring of *Pectinophora gossypiella* Cry1Ac resistance (**C**,**D**) in China. LC50 represents the median lethal concentration of the Cry1Ac protein to the collected larvae, and error bars stand for the 95% fiducial limits. The percent of cotton planted and bioassay data of LC50 from 2004-present are from Wan et al. [23].

## Data Availability

The data presented in this study are available on request from the corresponding author.

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
