# Peer review of "Managing Practical Resistance of Lepidopteran Pests to Bt Cotton in China"

_insects, 2023, doi:10.3390/insects14020179_

Round 1
Reviewer 1 Report
This is an exciting short review of resistance management in cotton insect pests in China. The authors clearly explain the 2 strategies that have been developed in China to stop the development of resistance in 2 important pests: Helicoverpa armigera and Pectinophora gossypiella. However, the review does not include a discussion of the possible mechanisms of action by which resistance has developed in both pests.
1. The authors explain the phenomenon of resistance from the point of view of classical genetics (homozygosity, heterozygosity, recessiveness, etc.), but there is no explanation related to the role of Cry and Vip toxin receptors in insects, which are only superficially mentioned in Table 1 (Cadherin, Aminopeptidase N). It is recommended to complete this type of explanation to better understand the role of these receptors, described almost as the only cause of resistance in insects.
2. It is also important to develop in a more complete way because it is considered that the introduction of Bt maize could affect the natural reservoirs of cotton pests since they do not feed on maize.
Author Response
This is an exciting short review of resistance management in cotton insect pests in China. The authors clearly explain the 2 strategies that have been developed in China to stop the development of resistance in 2 important pests: Helicoverpa armigera and Pectinophora gossypiella. However, the review does not include a discussion of the possible mechanisms of action by which resistance has developed in both pests.
- The authors explain the phenomenon of resistance from the point of view of classical genetics (homozygosity, heterozygosity, recessiveness, etc.), but there is no explanation related to the role of Cry and Vip toxin receptors in insects, which are only superficially mentioned in Table 1 (Cadherin, Aminopeptidase N). It is recommended to complete this type of explanation to better understand the role of these receptors, described almost as the only cause of resistance in insects.
Answer: Accepted. It has been added in this version (line 113 to 125).
- It is also important to develop in a more complete way because it is considered that the introduction of Bt maize could affect the natural reservoirs of cotton pests since they do not feed on maize.
Answer: Revised in lines of 360-362.
Reviewer 2 Report
The authors are to be commended on providing a comprehensive review of Bt cotton status, resistance concerns, for the key global pest; H. armigera and also PBW. I was impressed with the multiple years of data available for review, and that this was up to date re: current Bt proteins deployed in other countries as well. I have included Minor edits, to improve clarity and flow of the ms; the one main change for the Title, Abstract, is to use the term, Practical (vs Practice)....; as noted in margin of ms. This paper should have broad interest among researchers.
*See encl. file; all edits using Track changes; all Minor edits.

Author Response
The authors are to be commended on providing a comprehensive review of Bt cotton status, resistance concerns, for the key global pest; H. armigera and also PBW. I was impressed with the multiple years of data available for review, and that this was up to date re: current Bt proteins deployed in other countries as well. I have included Minor edits, to improve clarity and flow of the ms; the one main change for the Title, Abstract, is to use the term, Practical (vs Practice)....; as noted in margin of ms. This paper should have broad interest among researchers.
*See encl. file; all edits using Track changes; all Minor edits.
Answer: Many thanks. Accepted accordingly.
Commented [WDH1]: This is the term (spelled as defined by Tabashnik and colleagues, that I believe the authors refer to here, (and in the Title); thus my edits made. (I believe you have included the correct ref. citation to a Tabashnik paper; but authors should check, and cite at first use, in the Intro, and text below); and use practical throughout the ms.
Answer: Done.
Commented [WDH2]: Good to see this use of the correct term here; this is what I referred to on page 1, for Title of ms, and Abstract etc; use Practical throughout ms
Answer: Accepted.
Commented [WDH4]: Note to copy editors: there is a change in Front size. From 10.5 to 10.... from previous parag, to Line 258 (please check)
Answer: Accepted. It has been changed to size 10 now in this version.
Commented [WDH6]: Note: both citations need to be converted to correct format for Insects journal.
Answer: Accepted. We have added the references in follow of the correct format for Insects journal.